# Current Achievements and Future Prospects in Virus Elimination Technology for Functional Chrysanthemum

**DOI:** 10.3390/v15081770

**Published:** 2023-08-20

**Authors:** Kang Gao, Qingbing Chen, Bo Pan, Yahui Sun, Yuran Xu, Dongliang Chen, Hua Liu, Chang Luo, Xi Chen, Haiying Li, Conglin Huang

**Affiliations:** 1Institute of Grassland, Flowers and Ecology, Beijing Academy of Agriculture and Forestry Sciences, Beijing 100097, China; gaokang2015@bjfu.edu.cn (K.G.); dlchen1984@126.com (D.C.); liuhua1108@126.com (H.L.); changluo1983@126.com (C.L.); chengxiswzx@163.com (X.C.); 2College of Architecture, North China University of Water Resources and Electric Power, Zhengzhou 450046, China; chenqb1228@163.com (Q.C.); 17513367730@163.com (B.P.); sunyahui0713@163.com (Y.S.); xuyr1122@163.com (Y.X.)

**Keywords:** functional chrysanthemum, virus species, virus detection, virus elimination technology

## Abstract

Chrysanthemum is an important functional plant that is used for food, medicine and tea. Functional chrysanthemums become infected with viruses all around the world, seriously lowering their quality and yield. Viral infection has become an important limiting factor in chrysanthemum production. Functional chrysanthemum is often propagated asexually by cutting during production, and viral infection of seedlings is becoming increasingly serious. Chrysanthemums can be infected by a variety of viruses causing different symptoms. With the development of biotechnology, virus detection and virus-free technologies for chrysanthemum seedlings are becoming increasingly effective. In this study, the common virus species, virus detection methods and virus-free technology of chrysanthemum infection are reviewed to provide a theoretical basis for virus prevention, treatment and elimination in functional chrysanthemum.

## 1. Introduction

Chrysanthemum (*Chrysanthemum* × *morifolium* Ramat.), a perennial herb of the genus Chrysanthemum in the family Asteraceae, has a history of 4000 years in Chinese written records [1] and a cultivation history of more than 3000 years [2], with more than 3000 varieties [3]. Chrysanthemum is one of the top ten most famous flowers in China. With a graceful posture, bright colours and pleasant flowers, it has high ornamental value [4]. In addition, it also has edible, tea and medicinal functions [5,6]. “Functional chrysanthemum” is a general term for any edible, tea and medicinal chrysanthemum with nutritional and health functions [7]. This kind of chrysanthemum is rich in flavonoids, volatile oils, a variety of amino acids, chlorogenic acid, various trace elements and other chemical components [8,9,10,11]. It has a wide range of biological activities and can resist human immunodeficiency virus as well as possessing anti-inflammatory, anticancer, antioxidant, antimutation, and antitumour properties, providing cardiovascular protection, and so on [12,13,14,15,16,17,18,19]. It also has antibacterial, analgesic, and anti-fatigue effects, lowering blood sugar and regulating blood lipids [20,21,22,23]. With the improvement of living standards, people’s desire for better health is increasingly urgent, and the market demand for functional chrysanthemums is also growing daily. However, with the continuous expansion of the cultivation area for functional chrysanthemums, viral infection is common in the growth process [24,25,26,27,28,29], and chrysanthemums can be infected by a variety of viruses at the same time [30]. In addition, chrysanthemum is heterohexaploid [31,32], with a complex genetic background and high variability in hybrid offspring traits. Moreover, its self-incompatibility is very high, and long-term asexual reproduction leads to continuous accumulation of viruses [33,34,35]. This leads to germplasm degradation [24] and yield reduction [36,37,38] in various regions and increasingly serious phenomena such as the absence of buds or flowers, inferior colour and poor flower quality [39,40], which seriously hinder and limit the development of functional chrysanthemum industrialization [38,41,42,43]. There are problems of virus infection and virus removal in all plants during asexual reproduction, and the review of virus infection and virus removal techniques in chrysanthemum would lay a foundation for related research on other garden plants.

In this paper, the main viruses and viroids infecting functional chrysanthemums have been summarized from the aspects of virus species and transmission routes, and the current research progress in chrysanthemum virus detection and virus-free breeding technology are discussed to provide a reference for the effective prevention and control of functional chrysanthemum viral diseases.

## 2. Research Progress on Chrysanthemum Viral Diseases

### Main Virus Species Infecting Functional Chrysanthemum

Nine types of chrysanthemum viruses [19,28,44], namely, chrysanthemum virus B (CVB) [44,45], cucumber mosaic virus (CMV) [25], tomato mosaic virus (TMV) [25], tomato aspermy virus (TAV) [46], potato virus X (PVX) [46], potato virus Y (PVY) [47], tomato spotted wilt virus (TSWV) [48], chrysanthemum stunt viroid (CSVd) [49,50,51] and chrysanthemum chloritic mottle viroid (CChMVd) [52,53], have been reported in major functional chrysanthemum-producing areas. Most chrysanthemum viral diseases are quarantine hazards and can spread to a large area. Therefore, it is particularly important to investigate the infection status of functional chrysanthemum. On this basis, it is key to solve the problem of low yield and quality caused by chrysanthemum virus infection by using quick and effective detection methods to determine the virus species of chrysanthemum infection and to carry out elimination and rejuvenation of infected plants.

As shown in Table 1 and Figure 1, among the nine viruses, except for CMV and TAV, which belong to the same genus but different species, the other seven viruses belong to different species and genera. These viruses are mainly transmitted by aphid nonpersistent transmission, sap transmission, poisonous seed transmission, thrip transmission and cutting grafting, among which poisonous seeds can specifically spread CMV and thrips can specifically spread TSWV, causing harm to chrysanthemums. CSVd can only be spread by cutting grafting. In addition, except for CSVd, the other eight kinds of viruses can be transmitted by sap, seven kinds of viruses are mostly transmitted by aphids and sap, and two kinds of viruses are mostly transmitted by cuttings.

A functional chrysanthemum infected with a virus presents a variety of symptoms, as shown in Table 2 and Figure 2. According to the literature analysis, among the nine viruses reported in the main production areas of functional chrysanthemum, TSWV, CSVd and CChMVd have been the focus of relatively more studies on the symptoms of functional chrysanthemum infection, and the symptoms are relatively serious. Studies and reports on the other viruses after infection are relatively few.

According to the relevant literature, among the nine viruses reported in the main production areas of functional chrysanthemums, symptoms of TSWV, CSVd and CChMVd have been reported in functional chrysanthemum, which are relatively serious. Reports of the other six viruses after infection of functional chrysanthemum are relatively few.

When TSWV infects chrysanthemums, the leaves will exhibit yellow-brown ring spots (Figure 3A), the stems will exhibit black stripes (Figure 3J) and brown streaks (Figure 3L), the plants will be dwarfed (Figure 3O) or wilted (Figure 3M), and the infected plants will not produce flowers (Figure 3H). The most obvious symptom is upper stem rot (Figure 3K), leading to terminal wilt and death. The base of the shoot can be brown or black when the disease begins. The disease occasionally appears to have symptoms of leaf margin scorching, resulting in the death of seedlings. TSWV also causes leaf deformation, with some leaf edges showing obvious dark brown leaf spots (Figure 3N) and leaf spots more from the edge and developing inwards, which cause the whole leaf to die. A few functional chrysanthemums after infection showed mosaic leaves (Figure 3B), vein uplift (Figure 3C) and some abnormal flower colours (Figure 3G); the colour of the petals became light and yellow, petal distortion occurred (Figure 3I), and the number of petals decreased [75].

The main symptom of CSVd-infected chrysanthemums is dwarfism (Figure 3O), with plants that are 30~50% shorter than normal mature plants, and the infected chrysanthemums have small flower heads and changes in flower colour (Figure 3G). The flowering time can be more than ten days earlier, spots or markings (Figure 3D) appear on the leaves, and some varieties have seriously deformed (Figure 3F) leaves [76,77].

Chrysanthemum plants infected by CChMVd mainly show chlorosis, etiolation or spotted leaves (Figure 3E), which seriously affect the economic value and industrial development of functional chrysanthemums [25,78,79].

Virus-infected chrysanthemums mostly show negative infection or complex infection. Negative infection is asymptomatic early on, but the virus replicates normally in the plant. Complex infection is the simultaneous infection of multiple viruses, which makes the symptoms more complex [80,81]. Infection by these viruses seriously affects the commercial value of functional chrysanthemums and is not conducive to the promotion and large-scale production of chrysanthemums.

## 3. Research Progress in Chrysanthemum Virus Detection Technology

Plant viruses have been discovered over hundreds of years, and chrysanthemum virus B was discovered in the 1950s [82]. A variety of viruses infecting functional chrysanthemums were later found that greatly reduce the yield and quality of functional chrysanthemums. There are still no effective control measures, so it is particularly important to explore and establish a reasonable and effective elimination system. For the effective establishment of a virus-free system, the first step is to establish an efficient, convenient, and fast plant virus detection method. The following is an overview of the commonly used detection methods of functional chrysanthemums infected with viruses.

### 3.1. Indicator Plant Assay

The indicator plant detection method is to use the host to indicate that plants infected with one or more viruses and viroids will quickly show corresponding symptoms, so as to identify the virus [83]. Cai and Zhu selected *Nicotiana tabacum*, *Datura stramonium*, *Cucurbita pepo* and *Sativas cucumis* to identify the causal viruses. The results showed that the symptoms of plants inoculated with TAV were local necrotic spots and local chlorosis rings. In plants inoculated with CMV, the symptoms were mosaic, chlorosis and local dead spots [84]. Chen et al. took *Chenopodium quinoa*, *Nicotiana clevelandi*, *Nicotiana glutinosa*, *Tetragonia tetragonioides*, *Physalis floridana* and *Petunia hybrida* as indicator plants to identify the four varieties of chrysanthemum infected with the virus. The final results showed that ‘Jianyunxiang’, ‘Quanxiang Wansheng’ and ‘Quanxiang Chongtian’ were only infected with CVB, and ‘Chixian Jingou’ was only infected with TAV and CVB [85]. Supakitthanakorn et al. ground the chrysanthemum leaf that tested positive for TMV with 0.1 M phosphate buffer and used it to mechanically inoculate carborundum-dusted chrysanthemum leaves, confirming the pathogenicity of TMV in chrysanthemums [86].

The indicative plant detection method is characterized by its ease of operation and straightforward interpretation of results. However, it suffers from drawbacks such as a slow detection speed, susceptibility to experimental environment influences, and low sensitivity [87]. Additionally, chrysanthemums are susceptible to simultaneous infection by various viruses, making it highly challenging to discern the viral species based solely on disease symptoms.

### 3.2. Electron Microscope Observation

The electron microscopy observation method involves preparing samples from infected plant tissues and using short-wave electron beams to observe the structure of virus particles and the shape of virus capsid proteins under an electron microscope [16]. Due to its high resolution of up to 0.1 nm, while virus particle sizes range from 10 to 100 nm, the virus structure can be clearly observed. Currently, negative staining techniques are widely employed to process virus samples [88].

Kong and Li applied a mixture of virus sap and 1% glutaraldehyde onto a copper grid coated with a supporting film, followed by negative staining with 1% uranyl acetate, which led to the observation of spherical TAV virus particles [89]. Shu et al. diluted the virus sap, dropped the diluted solution onto a copper grid coated with Formvar membrane, and then negatively stained with 2% uranyl acetate, similarly observing a large number of spherical TAV virus particles [90]. Wu et al. conducted electron microscopy observation on chrysanthemums in the Kunming region. They found that the virus particle morphology exhibited four types: rod-shaped, filamentous, spherical, and near-linear, corresponding to the characteristics of TAV, CMV, TMV, and CVB. This indicates that chrysanthemums are commonly subjected to complex infections by multiple viruses [91]. And Supakitthanakorn et al. observed rigid rod particles approximately 200–300 nm in length under TEM in a naturally infected chrysanthemum leaf prepared by dipping and negative staining [86].

Transmission electron microscopy is expensive, requiring an understanding of virus morphology and characteristics. Additionally, results can sometimes be interfered with by fragmented organelles, contributing to its operational difficulty [92], which limits its applicability.

### 3.3. Enzyme-Linked Immunosorbent Assay

The enzyme-linked immunosorbent assay combines enzymes with antibodies through chemical methods to create enzyme-labeled antibodies. When enzyme-labeled antibodies encounter their corresponding substrates, they produce soluble or insoluble colored compounds.

Zhang et al. conducted virus detection on 158 chrysanthemum samples collected from the field using ELISA. Among them, 30 samples were found to contain CMV, with a detection rate of 19.0%, indicating a high infection rate of CMV in chrysanthemums [93]. Chen et al. performed virus detection on chrysanthemums from regions like Luoyuan and Xiamen in Fujian Province, revealing variations in the types of viruses infecting chrysanthemums in different areas. In Luoyuan, chrysanthemums were commonly infected with TAV, TMV, and PVY, and some with CMV. On the other hand, chrysanthemums in Xiamen were generally infected with TAV, PVX, and PVY [94]. Chen et al. used the double antibody sandwich method (DAS-ELISA) to detect viruses in chrysanthemum mother plants and in vitro-cultured shoots obtained from shoot-tip culture for virus elimination. They found that four chrysanthemum varieties were all infected with CVB, and one variety was also infected with TAV [85].

ELISA is highly specific and capable of testing a large number of samples. However, it has a slower testing speed and is prone to false-positive results. Therefore, for viruses with strong correlations, ELISA might not be suitable for accurate detection [95].

### 3.4. Molecular Biology Technology

Molecular biology can be used to confirm the presence of viruses by detecting viral nucleic acids (DNA or RNA) and is the most commonly used and easiest-to-perform method in scientific research experiments. It can detect Peake (pg) or even Feke (fg) grade [96] and can be used as an alternative to serological methods due to its higher accuracy and sensitivity. At present, the molecular biological technologies used in the detection and identification of chrysanthemum viruses mainly include nucleic acid hybridization, double-stranded RNA (dsRNA) electrophoresis, and reverse transcription polymerase chain reaction (RT-PCR). Among them, RT-PCR technology is the most widely used [97].

#### 3.4.1. Reverse Transcription Polymerase Chain Reaction (RT-PCR) and Its Related Detection Techniques

RT-PCR involves reverse transcribing the target RNA virus into cDNA and then amplifying it, resulting in the exponential amplification of minute viral nucleic acids. DNA viruses can be directly amplified using PCR. This technique achieves detection sensitivity at the Feke (fg) level [96], featuring high sensitivity and strong specificity. The analysis of infection status is often performed through gel electrophoresis [98]. RT-PCR is the most commonly used method in plant virus detection. It encompasses various approaches including single RT-PCR, multiplex RT-PCR, quantitative fluorescence PCR, and nested PCR [97].

Single RT-PCR refers to the extraction of RNA from tissue samples and reverse transcription into cDNA combined with a viral primer for RCR amplification to detect whether the plant contains a virus. However, this method can only be used to test for a single virus species. This technique was used to identify and detect TAV [98], TMV [86] and CVB [99] in functional chrysanthemums.

The operation process and reaction principle of multiple RT-PCR are no different from those of conventional PCR. Multiple viral primers can be added into a reaction system for reverse transcription with two or more target DNA or RNA regions for simultaneous detection of multiple PCR targets [100]. Because functional chrysanthemum plants under natural conditions are often infected by multiple viruses, multiple PCR techniques are often used to identify the species of viruses. Multiple RT–PCR technology can not only detect multiple viruses simultaneously but also has the advantages of high efficiency, less time consumption, less drug consumption, fast speed and high accuracy [101]. Song et al. used a multiplex PCR system to simultaneously detect TAV and CVB in infected chrysanthemums [102]. Liu established a multiplex RT-PCR system for the simultaneous detection of five viruses (CVB, TAV, CMV, TMV, and PVY) and two viroids (CChMVd and CSVd) infecting functional chrysanthemums. The experimental results showed that the system had high sensitivity and accuracy [103]. It is also proven that this method can be used for large-scale investigations of plant viruses and viroids [84].

Fluorescence quantitative RT-PCR (qRT) is a method that combines PCR technology with fluorescence detection. The principle is to use probes that can specifically bind PCR products and label fluorescent substances. The probes mainly include molecular beacon probes, hybridization probes and TaqMan fluorescence probes. At present, Taq DNA polymerase 5’ exonuclease activity is often used to degrade it in the reaction process, and the fluorescence groups are far away from the quenched groups to emit fluorescence to realize the monitoring of the whole process. Liu established a real-time PCR system for the detection of CVB, TAV, CMV, TMV, PVY, CChMVd and CSVd [103]. The seven species of chrysanthemum viruses and viroids can be detected efficiently and specifically even when the total RNA concentration is 100 fg/μL. This technique not only has high sensitivity and specificity but can also quantitatively detect the virus to avoid false positives, has stronger reliability, does not require gel electrophoresis, and avoids cross-contamination [104].

Common PCR techniques are able to detect very low contents of target genes, but in some cases, the virus content is extremely small, the target genes are unstable, and the product amplified by the primary primer still cannot be detected in gel electrophoresis, so a second round of PCR amplification based on the product of the first round of PCR is needed [21]. Nested PCR is a two-round PCR amplification reaction using two sets of PCR primers (namely, nested primers) [105]. The specificity of the amplified product is greatly improved. Guan et al. used nested PCR technology to detect CVB at up to 9–10 dilutions of cDNA obtained using gene-specific primers [106]. Compared with conventional RT-PCR, it showed higher sensitivity. Yan et al. used nested RT-PCR to detect both CVB and TAV in chrysanthemum [107], while Wu et al. used this technique to detect TAV, CVB, CMV, TMV and PVY in chrysanthemums at one time [108].

#### 3.4.2. Loop-Mediated Isothermal Amplification (LAMP)

LAMP technology uses the chain replacement activity of Bst DNA polymerase to provide the reaction power and completes the efficient, specific and sensitive amplification of target DNA in a short time under constant temperature conditions. Park et al. used LAMP technology to rapidly detect CChMVd at 65 °C [109]. Liu et al. designed four LAMP-specific primers that could identify six specific regions of the TAV CP sequence, which could quickly and accurately detect TAV in chrysanthemums and had advantages of higher sensitivity, speed and convenience than RT-PCR [110]. The colorimetric method for TMV developed in Supakitthanakorn’s study is the first of its kind and can be used for routine detection of TMV due to its speed, accuracy, sensitivity, and specificity [86].

Multiple RT-LAMP can detect multiple pathogens simultaneously in the same reaction system. Liu established a multielement RT-LAMP method for the simultaneous detection of CVB and CSVd infected chrysanthemum [103]. This method can simultaneously identify two pathogens in the same RT-LAMP reaction, thus improving the efficiency of simultaneous detection of these two chrysanthemum viral pathogens. Fukuta et al. designed LAMP primers for CMV and CVB CP genes and CSVd genomic RNA [111]. The established RT-LAMP method could simultaneously detect CMV, CVB and CSVd, and the results showed that the sensitivity of RT-LAMP was as high as that of RT-PCR for CSVd detection. However, the sensitivity was higher than that of CMV and CVB detected by RT-PCR.

As shown in Table 3, the current detection technologies are mainly divided into four categories: indicator plant assay, electron microscopy, ELISA and molecular biology, among which molecular biology has the most abundant types of technologies and the best detection speed and is the most widely used. At present, detectable viruses mainly include TAV, CVB, TMV, CMV, PVX and PVY, and the detectable viroids include CChMVd and CSVd. Among them, the detection methods for CVB are the most diverse, followed by TAV and CMV, and the detection methods for TSWV are the least diverse. For the viroids CChMVd and CSVd, molecular biological techniques are mainly used for detection. In addition, multiple RT-PCR and multiple RT-LAMP detection techniques are fast, accurate and efficient. They have excellent performance in the detection of chrysanthemum viruses and are thus the most widely used detection methods at present. The detection conditions of multiple RT-LAMP are simpler and more convenient than that of multiple RT-PCR and can be detected under constant temperature conditions, which also lays a foundation for the establishment of future on-site detection technology.

## 4. Research Progress on Chrysanthemum Virus Elimination Methods

To date, the chrysanthemum viruses and viroids reported to have been removed include CVB, CMV, TMV, TAV, CSVd and CChMVd. The main methods of virus elimination include shoot tip elimination, heat treatment combined with shoot tip elimination, antiviral treatment elimination combined with shoot tip elimination, ultralow temperature treatment, etc.

### 4.1. Shoot Tip Elimination

Viruses are unevenly distributed in the plant [112,113], being very low or absent in the shoot tip meristem [46]. Because there is no vascular bundle in the shoot tip meristem, the rate of cell division is higher than the rate of virus diffusion, thus inhibiting viral replication [114]. High concentrations of endogenous hormones in the shoot tip meristem also inhibit viral proliferation [69,115]. In principle, the smaller the length of the tip is, the better. If the shoot tip is too large, the elimination effect will be affected. A shoot tip that is too small is less likely to survive. The virus-free area of the shoot tip is generally no more than 0.5 mm. The most appropriate size of shoot tip for elimination can only be obtained by specific operations based on plant characteristics [116]. The elimination of chrysanthemum shoot tips mainly depends on a single action mode or compound action mode. Single mode of action: Kumar et al. successfully obtained virus-free seedlings in shoot tip cultures of chrysanthemum infected with both CMV and TAV [74]. Shoot tip elimination is often used in combination with other methods and is rarely used alone. In terms of the compound mode of action, Jeon et al. removed CSVd by heat treatment combined with shoot tip elimination [117]. Savitri et al. treated 3~5 cm of chrysanthemum shoot tips infected with CSVd at 4 °C for three months and then cultured them on medium containing 50 mg/L and 100 mg/L ribavirin. The virus-elimination rate reached 100% [118]. Ram et al. found that when CVB was removed from chrysanthemums, shoot tips with 0.3~1.0 mm and 2~3 leaf primordia were removed for culture, and the elimination rate of regenerated plants was 0 by RT-PCR detection, indicating that traditional elimination techniques have difficulty removing CVB, and there is an urgent need to establish an efficient CVB elimination method [119]. In a study by You et al., the virus-free rate of CVB-infected chrysanthemums was increased to 43.33% by secondary shoot tip culture [120]. In addition, Ram et al. cultured stem segments infected with CVB and incubated them at 38 °C for 30 days [119]. The regenerated plants were subjected to DAS-ELISA and RT-PCR, and the virus-free rates of CVB were 36.3% and 20%, respectively. After stem segments were treated with 40 mg/L medium containing 2-thiouracil for 25~30 d, the virus-free rates of CVB were 40% and 26.7%, respectively, by DAS-ELISA and RT-PCR. Hosokawa et al. successfully regenerated the shoot tip meristem of the leafless primordium attached to the root tip of chrysanthemums without CSVd, and 14.3% of the chrysanthemum seedlings were eliminated by PCR detection [121]. Shoot tip elimination is one of the most basic methods of functional chrysanthemum virus extraction.

### 4.2. Heat Treatment Elimination

Viruses and host plants have different high temperature tolerances, which makes the growth rate of plants at high temperature higher than the virus diffusion rate, and partial virus-free meristems can be obtained [122]. Heat treatment should be carried out with equipment such as a thermostat, which can treat the mother plant directly or the test-tube seedlings. Usually, the higher the temperature and the longer the duration, the better the elimination effect, but it is also easy to damage the plant [84]. Therefore, the appropriate time and temperature should be set according to the physiological condition of the plant. In practice, this method is often combined with the shoot tip elimination method to enhance the elimination effect. Zhao et al. treated chrysanthemum seedlings for 60 days and stripped 0.4~0.5 mm shoot tips for culture, and the TMV virus-free rate reached 94.1% [123]. Ram et al. found that the elimination rate of TAV was 51.7% by RT-PCR after chemical treatment and heat treatment [124]. When Zhao et al. used the chrysanthemum variety ‘Moju’ as the material to remove CVB, compared with the shoot tip culture method and ribavirin combined with the shoot tip culture method, heat treatment combined with the shoot tip culture method had the best effect, and ELISA results showed that the virus-free rate also reached 100% [123].

### 4.3. Antiviral Agent Treatment Elimination

At present, antiviral agents are mainly natural compounds and chemical agents. This method exploits the principle that antiviral agents prevent viral RNA from forming a cap structure in the triphosphate state and thus inhibit the virus. Commonly used antiviral agents include 5-dioxyuracil (DHT) and riboside triazolium (ribavirin) [125]. Among them, ribavirin is the most commonly used and can be injected directly or added to growth medium. It is often used in combination with shoot tip culture [116]. Budiarto et al. found that by using antiviral agents and heat treatment combined with shoot tip treatment of infected chrysanthemums, CVB could be completely removed, as verified by DAS-ELISA detection [126].

### 4.4. Ultralow-Temperature Treatment

Ultralow-temperature treatment is a method for treating virus-infected samples in liquid nitrogen for a short period of time to eradicate the virus, developed based on ultralow-temperature preservation [127]. It exploits the fact that mature vacuole cells contain more free water, which easily forms tiny ice grains, causing cell death under ultralow temperature conditions, while shoot tip meristem cells with fast division have dense cytoplasm, no mature vacuoles, and little free water, so they will not produce ice crystals and will survive at ultralow temperature conditions [127,128]. Wang obtained 21.9% virus-free regenerated plants by using hypothermia therapy for CVB elimination for the first time [12]. Jeon et al. incubated infected chrysanthemum at 4 °C for 4 weeks and then treated it at ultralow temperature, which effectively removed two kinds of viruses, CSVd and CChMVd, infecting chrysanthemums simultaneously [129].

Table 4 shows that the virus elimination rate was related to the size of the shoot tip culture. Different cultivars of chrysanthemum had the same size shoot tip requirements to remove the same virus, but the elimination rate was different. After the same virus-free method is applied to the same material, different detection techniques of detecting the virus-free rate will yield different results. RT-PCR is more sensitive than DAS-ELISA detection. Therefore, it is important to design different elimination and detection methods according to the specificity of different varieties.

In summary, the main elimination methods currently used are shoot tips, heat treatment, antiviral treatment and ultralow temperature. The most common is shoot tip elimination, but shoot tip virus elimination does not have a good effect by itself, so it is often combined with heat treatment technology and antiviral agent treatment, which can achieve better elimination effects. Among them, CVB and CMV can be completely removed by combining heat treatment with antiviral agents at shoot tip, and viroid CSVd can also be completely removed by using antiviral agents after low temperature treatment at 4 °C. These effective elimination methods are helpful to study the differences from before to after elimination from chrysanthemum plants, and to clarify the specific effects of CVB, CMV and CChMVd on the disease symptoms, physiological indexes and nutrient content of chrysanthemums. The existing elimination technology cannot effectively remove TAV, TMV and CChMVd, so much more research is still needed. At present, CVB, CMV, TAV, TMV, CSVd and CChMVd viruses have been more extensively studied in chrysanthemums, while PVX, PVY and TSWV have been studied less. The elimination technology of these three viruses still needs further research.

## 5. Research Progress on the Propagation of Virus-Free Chrysanthemum Seedlings

The propagation of chrysanthemums is mainly carried out by cutting, which is easy to popularize and inexpensive. However, in long-term production, due to virus accumulation, compound infection and other factors, plant dwarfing, nonflowering, yellow leaves and other problems greatly reduce the quality and value of chrysanthemums, affecting the development of the functional chrysanthemum industry [120,131]. Under natural conditions, once the virus invades the plant, it is difficult to eradicate, but tissue culture can effectively solve this problem [132], resulting in chrysanthemum elimination and rejuvenation [133]. Virus-free seedling breeding can be divided into several stages, such as the cultivation of sterile seedlings, propagation of rootless seedlings, rooting induction, cultivation and transplanting of seedlings, and evaluation of elimination effects.

### 5.1. Sterile Seedling Culture

MS has been used as the basic medium for the primary culture of chrysanthemums [134], and appropriate amounts of growth regulators such as 6-BA, NAA, KT or TDZ are added. In the process of tissue culture, there is a certain proportion of cytokinin and auxin in the medium. Generally, a higher concentration of cytokinin is conducive to the formation of adventitious buds, and a higher concentration of auxin is conducive to the formation of roots. When the concentration of cytokinin and auxin is in a certain equilibrium, it is conducive to the formation of calli [135,136].

Deng et al. established a rapid propagation technology system for virus-free ‘Chuzhou’ Chrysanthemum seedlings by using improved heat treatment combined with shoot tip meristem culture and concluded that the best medium for inducing calli from stem segments was MS + 2.0 mg/L 6-BA + 0.1 mg/L NAA. The optimal medium for shoot tip proliferation of ‘Chuzhou’ Chrysanthemum is MS + 1.0 mg/L 6-BA + 0.1 mg/L NAA [137]. Wang et al. used flower petals at the whitening stage as explants in the study of ‘Hangbai’ Chrysanthemums and found that the optimal culture medium for callus induction was MS + 0.2 mg/L NAA + 2 mg/L KT [138]. Deng and Wu studied tissue culture and rapid propagation techniques using chrysanthemum petals as explants and concluded that the best medium for inducing chrysanthemum calli was MS + 1.5 mg/L 6-BA + 0.1 mg/L NAA [139]. Liu et al. used the shoot tips of potted edible chrysanthemums as explants to induce callus formation and bud germination by regulating the hormone concentration and found that the optimal medium for callus differentiation and bud induction was MS + 1.0 mg/L 6-BA + 0.1 mg/L NAA, and the bud induction rate reached 55% [140].

### 5.2. Propagation of Virus-Free Seedlings

Secondary culture of functional chrysanthemums is an important method to obtain a large number of virus-free seedlings. One is to separate the callus from the bud and then inoculate the separated callus on the medium for culture to differentiate into more buds. The other is to cut the grown rootless test-tube seedlings into small segments with leaves or buds, inoculate them on the medium, and then grow them into seedlings [141]. When Hou et al. added 3.0 mg/L 6-BA and 0.01 mg/L NAA to MS base medium for chrysanthemum subculture, the shoots grew stronger, being suitable for chrysanthemum subculture expansion propagation [142]. Fu et al. used the terminal buds and axillary buds of the brain of chrysanthemums as tissue culture explants and concluded that the appropriate breeding medium for chrysanthemums was 1/2 MS + 0.5 mg/L 6-BA + 0.1 mg/L NAA [143].

### 5.3. Rooting Induction

Stable rooting is a necessary condition for transplanting cultivated seedlings. When auxin is present, most of the roots of tissue culture seedlings are stimulated and begin to grow [34]. IAA, IBA and NAA are common auxins, and different concentrations of NAA are added to MS medium in the rooting culture of chrysanthemum. Deng et al. showed that the optimal medium for the rooting culture of ‘Chuzhou’ Chrysanthemum was 1/2 MS + 0.1 mg/L NAA when studying virus-free seedlings [137]. Huang et al. concluded that the appropriate medium for inducing roots of ‘Chuzhou’ Chrysanthemum tissue culture seedlings was 1/2 MS + 0.2 mg/L IBA [144]. Fu et al. concluded that a medium suitable for inducing chrysanthemum roots was 1/2 MS + 0.1 mg/L NAA [143]. Yuan et al. concluded that the best medium for rooting induction was 1/2 MS + 0.1 mg/L NAA [145]. Liu et al. concluded that 1/2 MS + 1 mg/L IAA or 1/2 MS + 0.2 mg/L NAA medium was the most beneficial for rooting [146]. Deng et al. used 1/2 MS + 0.5 mg/L NAA as a rooting medium, and the rooting rate reached 100% [139].

### 5.4. Seedling Cultivation and Transplanting

Chrysanthemum virus-free seedlings are always grown in controlled indoor environments from inoculation to root growth. When tissue culture seedlings are removed from the tissue culture bottle, their growth environment changes greatly, and the external natural environment affects the survival rate of the seedlings. Therefore, it is necessary to cultivate seedlings before transplanting in the open field. The commonly used time to cultivate seedlings is when the rooting tissue culture seedlings are approximately 3 cm high, the white roots are exposed and the side roots are grown. Zhao et al. showed that 3 d was the most opportune time for seedling cultivation, when the survival rate of seedlings was the highest, reaching more than 95% [147]. If it was less or more than this amount of time, the survival rate decreased, and the seedling cultivation time was too short, so the seedlings could not adapt well to environmental changes. If the cultivation time is too long, the seedlings are prone to infection with bacterial viruses [148]. After seedling cultivation, the tissue culture seedlings are removed, the root residual medium is carefully washed with clean water, carbendazim or chlorothalene cleaning solution is used for sterilization for 1~2 min, and then the seedlings are transplanted into the substrate. The seedling cultivation substrate is generally silt soil and saprolite soil, the other is vermiculite, perlite and saprolite straw [149], and some transplanting substrate is a mixture of peat and coconut sugar [150]. The principle of matrix selection holds that tissue culture seedlings must quickly adapt to the external environment while transplanting the selection of good air permeability and high water retention performance of the matrix [147].

### 5.5. Evaluation of the Elimination Effect

Virus-free seedling propagation technology can effectively solve the problem of chrysanthemum viral infection, promote the growth and development of chrysanthemums from the root, and improve the fine quality of chrysanthemum plants and the content of effective chemical components [133]. Liu et al. found that the ratio of a high concentration of IAA hormone and a certain concentration of 6-BA was more conducive to promoting tissue callusing and improving the survival rate [146]. The plants treated with 1 mg/L IAA hormone had better growth, and the virus-free seedlings obtained from the treatment had strong plants and broad leaves, which could promote their growth and quality. Liu cultured ‘Hangbai’ Chrysanthemum explants in 1/2 MS medium with light 12 h/d, light intensity 25 μmol/(m^2^·s) and culture temperature (25 ± 2) °C [39]. The stem segments of virus-infected seedlings were cut off and inoculated in MS medium with 0.1 mg/L 6-BA and 0.1 mg/L NAA. When the plant height was approximately 3 cm, the plants were transferred to 1/2 MS medium with 0.1 mg/L NAA for rooting induction. When the root length was approximately 1 cm and the number of roots was approximately 4~5, the survival rate reached 100%. The branch number, chlorophyll, total flavonoid and chlorogenic acid contents of the virus-free seedlings were significantly higher than those of the conventional ‘Hangbai’ Chrysanthemum seedings. Wu et al. compared several shoot tip meristem culture methods. The results showed that the virus elimination rates of CVB and CSVd were 16.88%, 35.21% and 36.62%, respectively, when the shoot-tip meristem culture, heat treatment combined with shoot-tip meristem culture and ribavirin combined with shoot-tip meristem culture were used [151]. After inoculation of the virus-free vaccine into MS medium containing 1.0 mg/L 6-BA + 0.1 mg/L NAA, the highest survival rate of shoot tip meristem culture was 77.78%. It was found that the flower diameter, number of flowers per plant and yield of virus-free ‘Chuzhou’ Chrysanthemum seedlings were improved compared with non-virus-free seedlings. In addition, the chlorogenic acid content of virus-free seedlings was 9.0% higher than that of non-virus-free seedlings. The virus-free effect of functional chrysanthemum was evaluated mainly by phenotypic characteristics and nutrients, including plant height, stem diameter, root length, flower diameter and number of blooms. Physiological aspects include the chlorogenic acid content, flavonoid content and chlorophyll content.

In conclusion, different media should be used in different stages of tissue culture to achieve the best efficiency of clonal reproduction. In the culture propagation of rootless seedlings, the selection of explants and disinfection methods are different, and the hormone ratio of the callus-inducing medium is also different. KT and TDZ will also be used in relevant studies [152], and an MS medium with different concentrations of 6-BA and NAA is commonly used. In the rooting culture process, a 1/2 MS medium with different concentrations of NAA is commonly used. However, seedling cultivation and transplanting stages need to use different substrates to cultivate seedlings at different times according to the chrysanthemum variety. Evaluation of the detoxification effect has mainly been used in the comparative studies between virus-free and non-virus-free seedlings, and the phenotypic characteristics and nutrient content of virus-free seedlings have been significantly improved compared with those of non-virus-free seedlings.

## 6. Conclusions and Future Prospects

At present, nine kinds of chrysanthemum viruses, including CVB, CMV, TMV, TAV, PVX, PVY, TSWV, CSVd and CChMVD, have been reported in the main production areas of chrysanthemum [72]. The transmission modes include aphids, sap, virulent seeds, thrips, cutting and grafting. Seven of the viruses are mostly transmitted by sap, and the other two viruses are mostly transmitted by cutting grafting. In addition, all nine kinds of viruses affect the flowers, leaves, stems and whole plants of chrysanthemum. The leaves of chrysanthemum show the most serious effects, and the symptoms are mainly mottled leaves, necrosis spots, chlorosis and etiolation.

The current detection technologies are mainly divided into four categories: indicator plant assay, electron microscopy, ELISA and molecular biology, among which the molecular biology has the most different techniques, the highest detection speed and the highest sensitivity [153]. Multiple RT-PCR and RT-LAMP detection techniques are the most frequently used. At present, detectable viruses mainly include TAV, CVB, TMV, CMV, PVX and PVY, and the detectable viroids are CChMVd and CSVd. The detection technology of CVB is the most comprehensive, followed by TAV and CMV, and the detection of TSWV is the least comprehensive. The viroids CChMVd and CSVd are mainly detected by molecular biology.

The main methods of virus elimination include shoot tip culture, heat treatment, chemical treatment and ultralow temperature treatment. Shoot tip elimination is the most commonly used method of virus elimination, but the existing research shows that the combination of various methods can greatly improve the efficiency of virus elimination. For heat treatment combined with shoot tip elimination, the method of changing the temperature can be used for elimination. Due to the wide selection range of antiviral agents, the elimination effect from different plants is different, and it is necessary to consider the nature of the plant itself and the characteristics of the virus to avoid hindering the growth of the plant itself while improving the elimination rate. Research on ultralow-temperature elimination is still relatively inchoate, and further experimental research is needed.

During the propagation of sterile and virus-free seedlings, the selected explants and disinfection methods differ, as do the hormone ratios of the callus-inducing medium. An MS medium with different concentrations of 6-BA and NAA is commonly used, and KT and TDZ are also used in some studies [152]. In the rooting culture process, 1/2 MS medium with different concentrations of NAA is commonly used. The seedling refining and transplanting stages and the matrix ratios need to be determined for each variety. Evaluation of the detoxification effect is mainly used for comparative study between virus-free and virus seedlings. The phenotypic characteristics and nutrient content of virus-free seedlings are significantly improved compared with those of non-virus-free seedlings.

Chrysanthemum virus R (CVR) [154], zucchini yellow mosaic virus (ZYMV) [155] and chrysanthemum stem necrosis virus (CSNV) [156,157,158] have been found in some areas but have not been reported in the main production areas of functional chrysanthemum. It is necessary to carry out comprehensive research on chrysanthemum virus infection in the main production areas of functional chrysanthemum and identify the latest species of virus infecting chrysanthemums. The elimination of viroids is a challenge in the detoxification of functional chrysanthemums, and more effort should be made to detect and remove chrysanthemum viroids in the future. Early CSVd and CChMVd were concentrated on virus-free plants after detection [118,159]. Using in situ hybridization, Ebata et al. found that CChMVd infected the leaf primordia in the shoot tips meristem and stem meristem of chrysanthemum with a higher frequency, which can be used to study the elimination of chrysanthemum viroids from this perspective [160]. The symptoms caused by infection with a single virus in chrysanthemums are still unclear, so it is necessary to use advanced electron microscopy technologies to pinpoint the function of a single virus. The detection technology of functional chrysanthemum viral diseases is developing in the direction of high speed, sensitivity and specificity. Although the existing detection technologies can meet most virus identification needs, they are still a long way from the goal of convenience, efficiency and low cost. Digital PCR (dPCR) detection technology has been applied to the detection of a variety of plant viruses with high sensitivity [161,162,163] and can achieve multiple detection through different fluorescent channels [164]. In addition, in research on the breeding of virus-free chrysanthemum seedlings, there is still a lack of relevant research on whether virus-free seedlings will be infected again after continuous cultivation for multiple generations. On this basis, virus elimination from functional chrysanthemum seedlings should be carried out by the comprehensive use of different virus elimination techniques. With an in-depth understanding of the transmission route of plant viruses, breeding of excellent varieties with antiviral or virus resistance has become the most economical and effective control measure [165]. The continuous innovation of sequencing technology provides great convenience for finding target genes, and various resistant chrysanthemum varieties have emerged in an endless stream [166,167,168]. Currently, transgenic methods for antiviral chrysanthemums have been reported [169,170], and Torata et al. also cross-bred antiviral varieties to obtain new resistant varieties [171,172]. These findings provide a new direction for the much-needed study of nontoxic chrysanthemums.

## Figures and Tables

**Figure 1 viruses-15-01770-f001:**
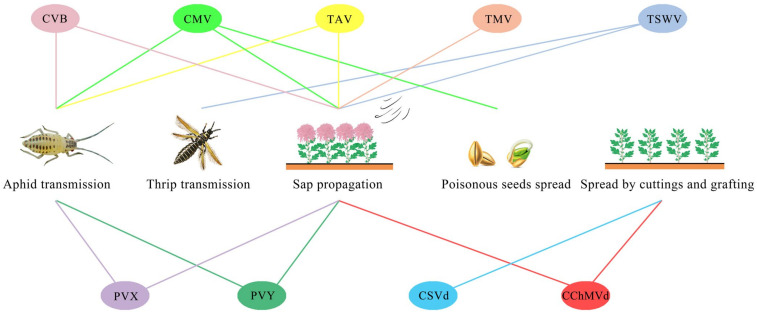
Diagram of chrysanthemum virus transmission.

**Figure 2 viruses-15-01770-f002:**
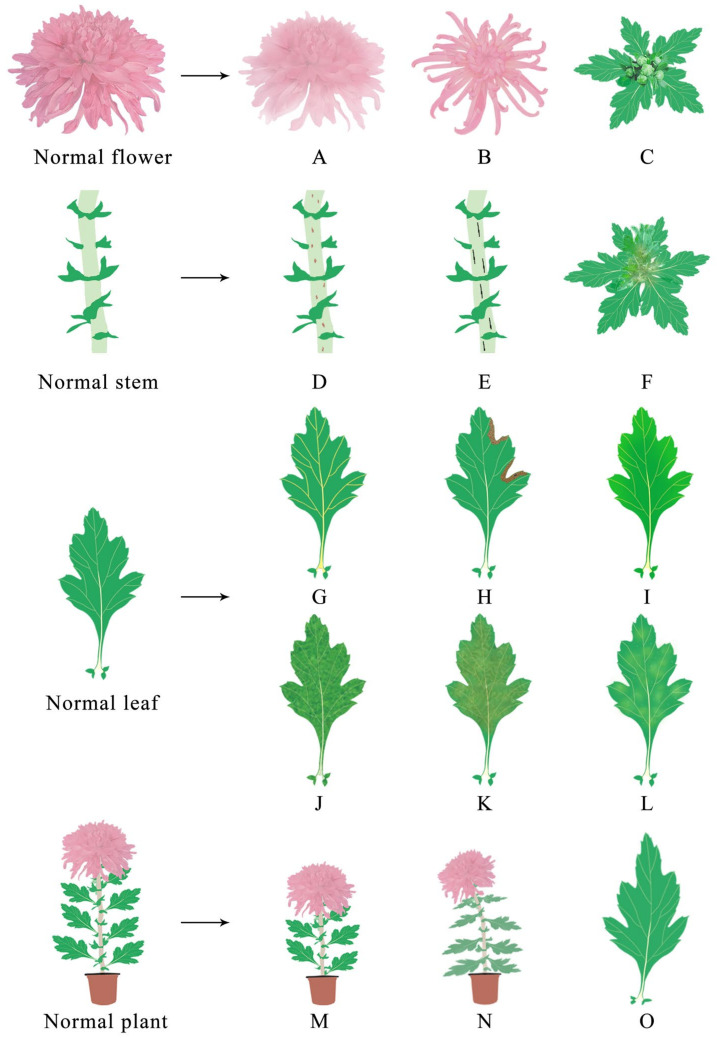
Diagram of functional chrysanthemum with virus infection symptoms. After infection with viruses, the flowers, stems, leaves and plants of the chrysanthemum will show different symptoms. Among them, the flower will display: (**A**) color changing; (**B**) floral deformity; (**C**) no flowers. The stems will display: (**D**) brown streaks; (**E**) black stripes; (**F**) upper stem rot. The leaves display: (**G**) vein uplift; (**H**) necrosis spots; (**I**) chlorisis and etiolation; (**J**) mottling; (**K**) yellow brown ring spots; (**L**) mosaicking; (**O**) deformation. The plants will display: (**M**) dwarfism; (**N**) wilt.

**Figure 3 viruses-15-01770-f003:**
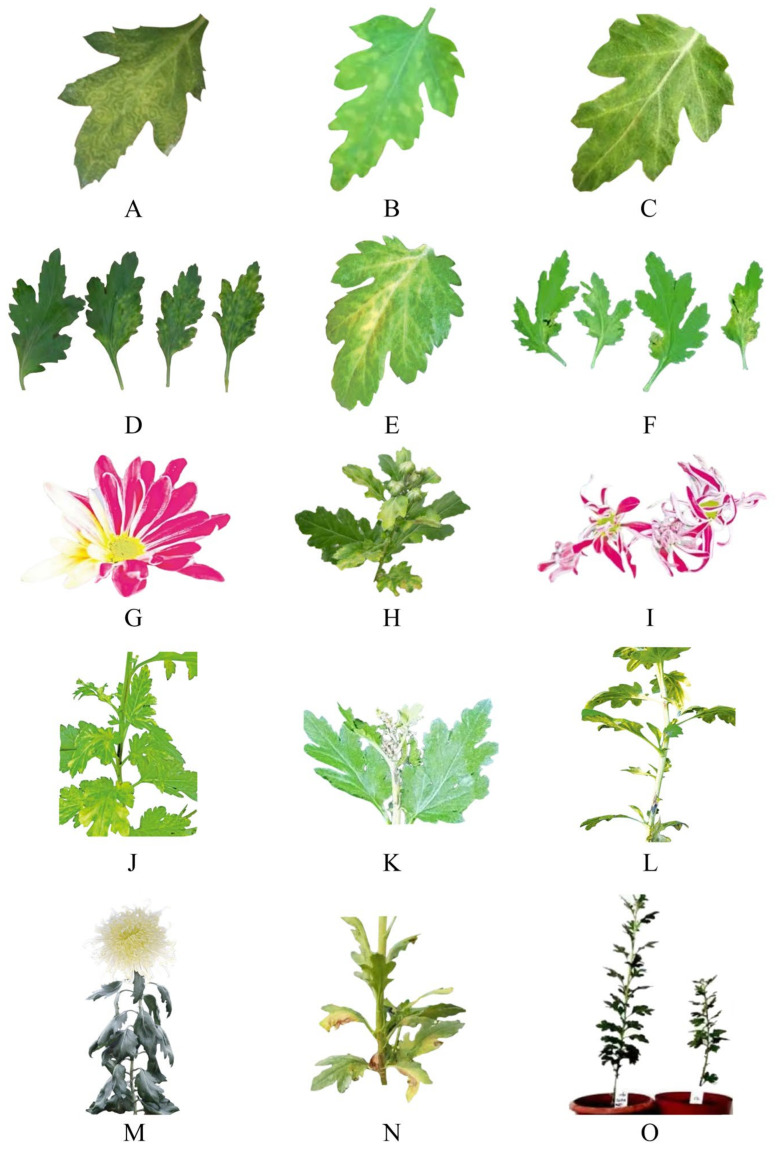
Symptoms of functional chrysanthemum infected with virus. They should be listed as: (**A**) yellow brown ring spot; (**B**) mosaic; (**C**) vein uplift; (**D**) mottled; (**E**) chlorosis and etiolation; (**F**) deformation; (**G**) color changing; (**H**) no flower; (**I**) floral deformity; (**J**) black stripe; (**K**) upper stem rot; (**L**) brown streaks; (**M**) wilt; (**N**) necrosis spot; (**O**) dwarfism.

**Table 1 viruses-15-01770-t001:** Analysis on the characteristics of nine viruses infecting functional chrysanthemums.

Virus Name	Genus	Mode of Transmission	Range of Infection	References
CVB	*Carlavirus*	Aphid and sap	Compositae	[54,55,56]
CMV	*Cucumovirus*	Aphid, sap and poisonous seed	Compositae and Legume	[57]
TAV	*Cucumovirus*	Aphid and sap	Compositae, Solanaceae and Chenopodiaceae	[36,58]
TMV	*Tobamovirus*	Sap	Compositae and Solanaceae	[59]
PVX	*Potexvirus*	Aphid and sap	Compositae, Solanaceae, and Legumes	[60]
PVY	*Potyvirus*	Aphid and sap	Compositae, Solanaceae, Legumes, and Quinoa	[61,62]
TSWV	*Orthotospovirus*	Sap and thrips	Compositae, Solanaceae, Legumes, and Cucurbitaceae	[47,63,64]
CSVd	*Pospiviroid*	Cuttage and grafting	Compositae, Solanaceae, and Aracanthaceae	[65]
CChMVd	*Pelamoviroid*	Cuttage, grafting and sap	Compositae	[66,67]

**Table 2 viruses-15-01770-t002:** Symptoms of virus infection in different parts of functional chrysanthemums.

Site of Infection	Virus Name	Symptom	References
Flower	CVB	Floral deformity	[68]
	TSWV	No flowers and color changing	[69]
	CSVd	Color changing and early flowering	[8,9]
Stem	TSWV	Brown streaks, black stripes and upper stem rot	[69]
Leaf	CMV	Vein uplift and mosaic	[59,70]
	TSWV	Yellow brown ring spots	[69]
	CVB	Mosaic and necrosis spots	[68]
	TAV	Mosaic, deformation and necrosis spots	[71]
	TMV	Mosaic, necrosis spots, mottled, chlorisis and etiolation	[72]
	PVX	Mosaic, necrosis spots, mottled, chlorosis and etiolation	[73]
	PVY	Mottled, chlorisis and etiolation	[59]
	CSVd	Deformation and mottled	[5,6]
	CChMVd	Chlorosis and etiolation	[72,74]
Plant	TAV	Dwarfism	[71]
	CSVd	Dwarfism	[4]
	TSWV	Dwarfism and wilt	[69]

**Table 3 viruses-15-01770-t003:** Comparison of detection techniques of chrysanthemum virus.

Detecting Techniques	Technological Type	The Species of Virus Detected	Detection Speed *	References
Indicator plant assay	——	TAV, CMV, TMV, CVB	+	[84,85]
Electron microscope observation	Negative staining technology	TAV, CMV, TMV, CVB	++	[86,89,90,91]
ELISA	——	TAV, CMV, TMV, PVX, PVY, CVB	++	[85,93,94]
Molecular Biology Technology	Single RT-PCR	TAV, TMV, CVB	+++	[86,98,99]
	Multiple RT-PCR	TAV, CVB, CMV, TMV, PVY, CChMVd, CSVd	++++	[102,103]
	Fluorescence quantitative RT-PCR	TAV, CVB, CMV, TMV, PVY, CChMVd, CSVd	+++	[103]
	Nest-PCR	TAV, CVB, CMV, TMV, PVY	+++	[107,108]
	RT-LAMP	TAV, TMV, CChMVd	+++	[86,109,110]
	Multiple RT-LAMP	CVB, CMV, CSVd	++++	[103,111]

* Detection speed is graded according to the number of “+”.

**Table 4 viruses-15-01770-t004:** Main virus elimination methods of chrysanthemum.

Virus Name	Elimination Method	Shoot Tip Size (mm)	Rate of Elimination (%)	Detection Techniques	References
CVB	Shoot tip	0.3~1	0	RT-PCR	[119]
	Shoot tip	<0.2	43.6	DAS-ELISA	[126]
	Shoot tip	0.3~0.7	43.33	RT-PCR	[120]
	Shoot tip with heat treatment	0.3~1	20	RT-PCR	[119]
			36.3	DAS-ELISA	
	Shoot tip with heat treatment	0.4~0.5	100	ELISA	[123]
	Shoot tip, antiviral agents and heat treatment	<0.2	100	DAS-ELISA	[126]
	Ultralow-temperature	2.0	21.9	RT-PCR	[12]
			24.4	DAS-ELISA	
	Antiviral agents	0.3~1	26.7	RT-PCR	[119]
			40	DAS-ELISA	
CMV	Shoot tip	0.3	72	RT-PCR	[130]
			84	DAS-ELISA	
	Shoot tip	0.3	65.6	RT-PCR	[74]
			78.1	DAS-ELISA	
	Shoot tip with heat treatment	0.4~0.5	100	ELISA	[84]
TAV	Shoot tip	0.3	65.6	RT-PCR	[74]
			78.1	DAS-ELISA	
TMV	Shoot tip, antiviral agents and heat treatment	0.3~0.4	51.7	RT-PCR	[124]
			65.9	ELISA	
CSVd	Shoot tip	0.3	14.3	Nest PCR	[121]
	Antiviral agents with 4 °C treatment	30~50	100	RT-PCR	[118]
	Ultralow-temperature	30~50	20	Nest PCR	[129]
CChMVd	Ultralow-temperature	30~50	20	Nest PCR	[129]

## Data Availability

Not applicable.

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
