# Peer review of "Current Achievements and Future Prospects in Virus Elimination Technology for Functional Chrysanthemum"

_viruses, 2023, doi:10.3390/v15081770_

Round 1

Reviewer 1 Report

The manuscript mainly focused on the summary of viruses and viroids that infect functional chrysanthemums from the aspects of virus species, transmission routes, and harm and discussed the current research progress of chrysanthemum virus detection and virus-free breeding technology, which provides a reference for the effective prevention and control of functional chrysanthemum viral diseases. From a personal perspective, I think it is a meaningful study that gave us comprehensive knowledge about the prevention and treatment of viral infections of chrysanthemum plants. These informations are vital in assisting farmers in cultivating chrysanthemums healthily. However, I am afraid that a major revision is needed due to the incorrect citation of some references, incomplete data, and poor writing and organization of the text. Some major/minor comments can be found below.

1. Check the accuracy of the cited references, especially those about the CVB.

2. Another major issue with the manuscript is related to the quality of the English. On first reviewing the paper, I made numerous edits/changes to the text and found myself unnecessarily distracted from assessing the paper on scientific grounds. There is no doubt that the quality of the paper is affected.

3. The abbreviation should be used uniformly.

4. Scientific names should be written in italics.

5. There are a lot of issues with spacing, punctuation, capitalization, etc, in the manuscript.

6. Please uniform the writing of Figure or Fig in the text.

7. The tables are poorly formatted, and they should be improved.

8. The page number is incorrect.

9. I was surprised to see that Table 3 directly used the header of the submission template information. Thus the manuscript needs to be carefully checked and corrected.

10. Please check all the references and ensure they conform to acceptable formats of Viruses.

Extensive editing of English language required.

Author Response

Response to Reviewer 1 Comments

Response letter

(Manuscript ID:viruses-2531288)

Dear Reviewer of Viruses,

First of all, I would like to thank the editors and reviewers for your valuable suggestions, which are crucial to the improvement and promotion of this article. We have made careful revisions according to your suggestions and replied point-to-point. All revised contents have been marked in yellow and green in the marked version of the manuscript. In addition to the 10 suggestions you put forward, I have also proofread and modified other contents of the manuscript. Submissions include new figures and tables, clear version , highlighted version and polished version. However, due to system reasons, the attachment can only upload one file, which is the modified file of the highlighted version, in which the yellow part is the modified part, and the green part with the delete line is the part that needs to be deleted.

I hope this revision will meet the high standard of Viruses.

Sincerely Yours,

Dr. Chen

E-Mail: [email protected]

Response to Reviewer 1 Comments:

The manuscript mainly focused on the summary of viruses and viroids that infect functional chrysanthemums from the aspects of virus species, transmission routes, and harm and discussed the current research progress of chrysanthemum virus detection and virus-free breeding technology, which provides a reference for the effective prevention and control of functional chrysanthemum viral diseases. From a personal perspective, I think it is a meaningful study that gave us comprehensive knowledge about the prevention and treatment of viral infections of chrysanthemum plants. These informations are vital in assisting farmers in cultivating chrysanthemums healthily. However, I am afraid that a major revision is needed due to the incorrect citation of some references, incomplete data, and poor writing and organization of the text.

Response: I thank for your suggestion. I have fully agree with your comment and use yellow and green color to make relevant changes in the revised manuscript. Below are point-to-point responses to specific comments

Point 1: Check the accuracy of the cited references, especially those about the CVB.

Response 1: Thank you for your suggestion. In the revised version, the incorrect literature citations have been removed, and the relevant literature on CVB cited has been carefully checked.

Point 2: Another major issue with the manuscript is related to the quality of the English. On first reviewing the paper, I made numerous edits/changes to the text and found myself unnecessarily distracted from assessing the paper on scientific grounds. There is no doubt that the quality of the paper is affected. 

Response 2: Thank you for your valuable advice, which is very important for the improvement of the article. The specific language problems have been modified. For other problems, we have found a professional polishing agency, see the attached version after polishing. In the marked version, lines 150 to 235, 340 to 342, 406 to 422, 471 to 484, 531 to 534, 555 to 589, 597 to 601, 608 to 610, and 617 to 623 have been carefully modified to make the full text more scientific.

Point 3: The abbreviation should be used uniformly. 

Response 3: Thank you for your comments. In the markup version, I have made uniform changes to the abbreviations for the virus section on lines 62 to 66, 397, and 590 to 591.

Point 4: Scientific names should be written in italics.

Response 4: Thank you for your suggestions on the details. In the markup version, I have changed the scientific names in italics on lines 27, 153 to 154 and 157 to 159.

Point 5: There are a lot of issues with spacing, punctuation, capitalization, etc, in the manuscript.

Response 5: Thank you for your valuable advice. In the markup version, I have made changes to the Spaces, punctuation, and capitalization of lines 12 through 630 and tables 1, 2, 3, and 4. These suggestions are crucial to improving the format of the manuscript.

Point 6: Please uniform the writing of Figure or Fig in the text.

Response 6: For this problem, I've changed all of them to the full name "Figure," specifically on lines 113 through 131 in the markup version.

Point 7: The tables are poorly formatted, and they should be improved.

Response 7: Thank you for your suggestion. A standardized and beautiful table will help improve the overall quality of the manuscript. In the modified version, I changed the space, punctuation, and capitalization issues for tables 1, 2, 3, and 4. In addition, for table 2, I also modified the arrangement order of the table and the order of references, so that the table is more accurate and beautiful, and can correspond to the arrangement order of Figure 2. As for Table 3, according to your suggestion and relevant literature review, it is considered that "Detection effect" cannot be measured by orders of magnitude, so this column is deleted. And add the "References" column to make Table 3 more scientific. For table 4, the references were first adjusted, and then the box of "CChMVd, CSVd" was rearranged to make the table more reasonable and beautiful.

Point 8: The page number is incorrect.

Response 8: Thank you for your detailed suggestion. I have rearranged the full page numbers, which are located in the upper right corner of the tagged version of this manuscript.

Point 9: I was surprised to see that Table 3 directly used the header of the submission template information. Thus the manuscript needs to be carefully checked and corrected.

Response 9: Thank you for your advice and sorry for my carelessness. I have renamed the table questions according to the contents of Table 3, the related error messages have been deleted, and other similar problems in the whole paper have been carefully modified.

Point 10: Please check all the references and ensure they conform to acceptable formats of Viruses.

Response 10: Thank you for your suggestions to make the manuscript more responsive to the requirements of Viruses. All references have been modified according to the submission format of the Viruses.This is in line 633 through 979 of the markup version.

Reviewer 2 Report

Please find the attached a file. My comments are in PDF file. 

I found several uncomfortable sentences. I already commented in a PDF file. Please find it. 

Author Response

Response to Reviewer 2 Comments

Response letter

(Manuscript ID:viruses-2531288)

Dear Reviewer of Viruses,

First of all, I would like to thank the editors and reviewers for your valuable suggestions, which are crucial to the improvement and promotion of this article. We have made careful revisions according to your suggestions and replied point-to-point. All revised contents have been marked in yellow and green in the marked version of the manuscript. In addition to your suggestions, I have also proofread and revised other contents of the manuscript. Submissions include new figures and tables, clear version , highlighted version and polished version. However, due to system reasons, the attachment can only upload one file, which is the modified file of the highlighted version, in which the yellow part is the modified part, and the green part with the delete line is the part that needs to be deleted.

I hope this revision will meet the high standard of Viruses.

Sincerely Yours,

Dr. Chen

E-Mail: [email protected]

Response to Reviewer 2 Comments:

Reviewer specific comments in PDF file

Response: I thank for your suggestion. I have fully agree with your comment and use yellow and green color to make relevant changes in the revised manuscript. Below are point-to-point responses to specific comments.

Point 1: Need space between etc. and references. I saw many mistakes in this manuscript. Please revise them.The first capital letter should be lowercase letters. In general, in case of common n

ame of viruses, the name should be non-italic and lowercase letters. (Format problems)

Response 1: Thank you for your valuable advice. In the markup version, I have made changes to the Spaces, punctuation, and capitalization of lines 12 through 630 and tables 1, 2, 3, and 4. These suggestions are crucial to improving the format of the manuscript. And I have changed the scientific names in italics on lines 27, 153 to 154 and 157 to 159.

Point 2: Please carefully revise it. I understand the authors would like to introduce virus detection techniques. However, indicator plant assay, TEM assay, and ELISA assay are pretty old fashions. Please shortly present them. Instead of them, the authors can introduce more recent modern techniques for virus detections such as isothermal amplification and digital PCR etc. And some paragraphs need to be shortened and some sentences need to be improved. Is it correct on advantages of ELISA? Is it proper expression? Recommend "elimination or eradication". (Content problems)

Response 2: Thank you for your valuable advice, which is very important for the improvement of the article. The specific language problems have been modified. For other problems, we have found a professional polishing agency, see the attached version after polishing. In the marked version, lines 150 to 235, 318 to 342, 406 to 422, 471 to 484, 531 to 534, 555 to 589, 597 to 601, 608 to 610, and 617 to 623 have been carefully modified to make the full text more scientific. I have made uniform changes to the abbreviations for the virus section on lines 62 to 66, 397, and 590 to 591.

Point 3: How the authors evaluate "Detection effect"? Pls. add references. Please provide description of Fig. 2. Please carefully double-check on legend of Fig. 3. Need space between (a) Yellow brown ring spot etc. Image is not clear. Please provide more high resolution images. (Figure and table problems)

Response 3: Thank you for your suggestion. A standardized and beautiful table will help improve the overall quality of the manuscript. In the modified version, I changed the space, punctuation, and capitalization issues for tables 1, 2, 3, and 4. In addition, for table 2, I also modified the arrangement order of the table and the order of references, so that the table is more accurate and beautiful, and can correspond to the arrangement order of Figure 2. As for Table 3, according to your suggestion and relevant literature review, it is considered that "Detection effect" cannot be measured by orders of magnitude, so this column is deleted. And add the "References" column to make Table 3 more scientific. For table 4, the references were first adjusted, and then the box of "CChMVd, CSVd" was rearranged to make the table more reasonable and beautiful. The sharpness of Picture 1 and picture 2 has been adjusted to 300dpi and the description of Picture 2 has been added in lines 96 to 101.

Point 4: Please revise "Ref" correctly. No journal name. Please provide or discuss on updated powerful detection methods and elimination techniques. Please provide "HTS" or "Nanopore sequencing" based approaches for virus detection in detail. Please double-check ref. 30. Can't find it. (Reference problems)

Response 4: Thank you for your suggestions to make the manuscript more responsive to the requirements of Viruses. All references have been modified according to the submission format of the Viruses.This is in line 633 through 979 of the markup version. And added the relevant literature in the Discussion and Outlook section 165 to 170, in lines 952 to 963. Regarding the specific clarification of "HTS" or "Nanopore sequencing", after re-checking the relevant literature, it is decided to delete these two references and add more appropriate relevant documents. I am sorry for marking document 30 as 31 due to my carelessness, and have corrected this problem.

Round 2

Reviewer 1 Report

 I recommend to accept the manuscript for publication.

Reviewer 2 Report

The authors have concisely revised based on reviewer's comments.